# Return on Investment (ROI) of Three Vaccination Programmes in Italy: HPV at 12 Years, Herpes Zoster in Adults, and Influenza in the Elderly

**DOI:** 10.3390/vaccines11050924

**Published:** 2023-04-30

**Authors:** Marco Barbieri, Sara Boccalini

**Affiliations:** 1York Health Economics Consortium, York YO10 5NQ, UK; 2Sezione di Igiene, Medicina Preventiva, Infermieristica e Sanità Pubblica, Università degli Studi di Firenze, 50121 Firenze, Italy

**Keywords:** HPV, herpes zoster, flu, models, vaccination, ROI

## Abstract

The calculation of the return on investment (ROI) allows the estimation of the opportunity cost of a series of interventions and can therefore help to make allocative choices. The objective of this study is to estimate the ROI of three vaccinations (HPV for adolescents, HZ for adults, and influenza for the elderly) in the Italian context, considering the impact of increasing vaccination coverage based on target objectives of the National Immunization Plan (PNPV) 2017–2019 and accounting for different eligibility criteria of each vaccination. Three separate static cohort models were constructed, including the eligible population for these vaccinations on the basis of the PNPV 2017–2019 and following this population until death (lifetime horizon) or until vaccination waning. Each model compares the level of investment at current vaccine coverage rates (current VCRs scenario) with that of optimal NIP target VCRs with a non-vaccination scenario. The ROI for HPV vaccination was the highest among the programs compared and was always above 1 (range: 1.4–3.58), while lower values were estimated for influenza vaccination in the elderly (range 0.48–0.53) and for vaccination against HZ (range: 0.09–0.27). Our analysis showed that a significant proportion of savings generated by vaccination programs occurred outside the NHS perspective and might often not be estimated with other forms of economic evaluation.

## 1. Introduction 

In assessing the value of vaccination programmes, some health economists have argued that classical economic evaluations, commonly used for other health interventions, may not take into account all relevant elements related to these programmes [1,2]. Indeed, it should be emphasised that social, ethical, and economic impacts of vaccination programmes also affect society at large, and not only the vaccinated individual and the healthcare system [3]. In particular, the impact of vaccination programmes on long-term economic and macroeconomic behaviour is very important, but difficult to quantify [4].

Return-on-investment (ROI) analysis has been initially applied in the area of vaccine prevention led by the Johns Hopkins Bloomberg School of Public Health, with the support of the Global Alliance for Vaccine Immunization (GAVI) [5]. It consists in a set of evaluations to support/demonstrate the sustainability and profitability of the implementation of new vaccine programmes in those contexts where they were absent (i.e., in developing countries). According to the most recent evaluations [6] the introduction of 10 paediatric vaccinations in 73 developing countries would produce a ROI over a decade ranging from an average of $21 per dollar invested, if the more conservative cost of illness (COI) methodology is used, to around $54 per dollar invested if Value of Statistical life (VRS) evaluations are used [7].

However, ROI is receiving increasing attention also in more advanced economic contexts: a number of authors have carried out a review of the ROIs of several public interventions, including education, taxation, and public health [8]. 

It should be emphasised that the ROI analysis presented in this study, unlike cost-effectiveness analyses, does not consider a clinical outcome that directly takes into account the impact of the vaccine programme analysed (e.g., quality-adjusted life-year gained, QALYs) and therefore focuses exclusively on the economic elements of the investment. On the other hand, however, it has some clear advantages, such as the inclusion of costs often not considered in standard cost-effectiveness evaluations, the possibility of evaluating a set of vaccinations at the same time, and the relevance for society at large and not only for the National Health Service (NHS), as is often the case with cost-effectiveness analyses. For these reasons, it can be suggested that the ROI analysis should be regarded as complementary and not as a substitute for cost-effectiveness analyses. 

This analysis was inspired by a recent exercise conducted by the British Office of Health Economics (OHE) [9] and follows the methodological approach of the handbook of immunization economics from Harvard’s DOVE initiative [10].

The aim of the present study, therefore, was to calculate the ROI of three different vaccination programmes in Italy: vaccination against the Human Papilloma Virus (HPV) in 12-year-old girls and boys, against Herpes Zoster (HZ) for adults, and against influenza for the elderly (≥65 years).

## 2. Methods

Three separate ex-novo static cohort models were built for the three vaccination programmes considered. Specifically, we considered a fixed population entering the model in the year 2020 that was followed over time. The study population was taken from Istat Demo data on the first of January 2020 [11], taking into account (or accounting for) the age at which individuals are eligible for the vaccination under consideration on the basis of the National Vaccine Prevention Plan (PNPV) 2017–2019 (still in place in Italy) [12]. Individuals were followed until death (lifetime horizon) or until vaccination waning, depending on the programme considered. 

Specifically, the three vaccination programmes considered in this study were as follows: the universal (male and female) HPV vaccination with nono-valent vaccine for 12-year-olds, actively offered since 2017 in all regions;the herpes zoster (HZ) vaccination according to the current indications of the PNPV 2017–2019: the cohort of subjects = 65 years of age, and subjects aged >50 years at increased risk (diabetes, chronic obstructive pulmonary disease (COPD), heart failure, subjects on immunosuppressive therapy);annual influenza vaccination in persons ≥ 65 years of age.

The choice of these three vaccinations was based on their heterogeneity: immunisation against HPV concerns adolescents and protects them against clinical disease and complications that appear over their lifetime, the Herpes-Zoster vaccine protects against an individual risk of disease and it is also recommended for distinct adult subpopulations according to the increased risk of developing HZ, while the influenza vaccination has the peculiarity of being repeated annually. The evaluation of the different ROIs of these three vaccination programmes might highlight important features for decision-makers. 

Each model compares the level of investment based on current vaccination coverage (current VCRs) to the investment based on the 2017–2019 PNPV optimal vaccination targets (optimal target scenario). Finally, both scenarios were compared to a hypothetical ‘non-vaccination’ (NV) scenario. 

In general, each model estimates the number of expected cases (e.g., incident cases of HZ or influenza) if the individuals included in the starting cohort were not vaccinated, based on the historical incidence rate or the literature data of the infections considered. Each disease considered is associated with a risk of complications with possible hospitalisation and/or loss of productivity. The number of averted cases associated with each vaccination considered is then calculated based on the vaccine efficacy value and coverage rates assumed. Accordingly, a certain degree of protection and coverage rates decreases the number of complications. Finally, the costs associated with each complication considered are calculated. The analysis was conducted from a societal perspective, including both direct costs associated with disease episodes and indirect costs associated with productivity lost. A discount rate of 3% was applied to future costs. 

The ROI was calculated according to the following formula: ROI = (Δ DIRECT AND INDIRECT COSTS DUE TO THE DISEASE)/(Δ VACCINATION COST) 

A ROI value greater than 1 indicates a situation in which the savings in terms of direct and indirect costs of a vaccination strategy are greater than the costs of the vaccination programme; in other words, savings due to the reduction in future clinical cases and complications are higher than the costs of vaccination. A value of less than 1, on the other hand, suggests that savings are lower than the cost of the investment (which might still be recommendable in terms of cost-effectiveness, although not dominant).

It should be noted that in contrast to a cost-benefit analysis, in which there is an attempt to quantify in monetary terms the clinical benefits by estimating the willingness to pay of individuals, in this ROI estimation there is no monetary calculation of clinical benefits, but the change in direct or indirect costs due to the disease are directly quantified in the reduction in (for example) hospitalisations or other healthcare resources. The disadvantage of this approach is that it does not directly consider the clinical benefits of a health programme, but the advantage over cost-benefit analysis lies in avoiding the quantitative estimation and ethical difficulties of monetising a gain in quality of life and/or survival. 

### 2.1. The HPV Model 

The human papillomavirus virus (HPV) is responsible for the most frequent sexually transmitted infection in the world (Koutsky, 1997; Baseman and Koutsky, 2005) [13,14]. The HPV vaccine is therefore offered to adolescents before sexual debut and to mid-adulthood persons at increased risk of HPV infection.

The ROI model considers a population eligible for vaccination in the 12th year of life in 2020, using ISTAT Demo data (295,927 girls and 278,434 boys resident in Italy on the first of January 2020), for whom a target vaccination target of 95% is set by the PNPV 2017–19. In the ‘non-vaccination’ scenario we applied age-specific HPV-related hospitalisation rates found in the literature [15] and observed in the Italian population that had an hospital discharge (SDO) between 2006 and 2018, a period of observation that we can assume interests persons who did not benefit from HPV vaccination which, in Italy, interests ladies born after 1994 and who are still too young to show the benefits on immunization. For the ‘current coverage’ and ‘target coverage’ scenarios we assumed a reduction in cases by applying vaccine efficacy to the proportion of vaccinated subjects. The latest recorded vaccination coverages for the primary cohort refers to the 2007 cohort: 41.6% of vaccinated females with a full cycle in 2019 and 32.2% of males [16]. 

Vaccination is routinely offered at the age of 12 years, whereas HPV-related diseases prevented can appear decades after HPV vaccination: thus, the time horizon of the model is lifetime. 

The analysis applies a discount rate of 3% to direct and indirect costs, distinguishing the discount horizon between women and men according to life expectancy at 12 years: 73 years for women, 69 for men [17], respectively. 

#### 2.1.1. Epidemiological and Vaccine Efficacy Data 

HPV vaccination is indicated in the prevention of pre-cancerous lesions and cancers affecting the cervix, vulva, vagina, penis, and anus, as well as in the prevention of genital condylomata (Condyloma acuminata) caused by the HPV subtypes contained in the vaccine. The probability of developing an HPV-related event over the lifetime horizon was calculated from the age-specific hospitalisation rates observed in the SDO, aggregated for the period 2006–2018, resulting in cumulative or lifetime incidence rates. 

Vaccine efficacy was evaluated on the basis of data from clinical trials of the 4- and 9-valent vaccine [18,19]. The duration of protection was assumed to be lifetime with respect to HPV strains 6/11/16/18/31/33/45/52/58. In the model, 9-valent vaccine efficacy is applied to incident HPV-related conditions, taking into account: (i) the fraction of each disease attributable to HPV infection and; (ii) the fraction attributable to the 9 vaccine serotypes currently included in the 9-valent vaccine, according to the latest European data on the HPV9 attributable fraction. The number of avoided cases was therefore calculated considering vaccine efficacy by specific HPV-related condition and the fraction of vaccinated subjects (Table 1).

To date, HPV vaccination with a 9-valent vaccine is not indicated in Italy in the prevention oro-pharyngeal cancer. However, a fraction of these cancers is known to be attributable to the HPV virus, and the Food and Drug Administration (FDA) has recently approved the extension of indication in the USA based on efficacy data in the prevention of HPV-related anogenital lesions. In the model we assumed a vaccine efficacy value for the fraction of preventable oro-pharyngeal cancers similarly to the vaccine efficacy values inferred by FDA to extend the indication [20].

**Table 1 vaccines-11-00924-t001:** HPV vaccination efficacy data included in the HPV model.

HPV-Related Condition	Efficacy HPV9 from RCP	HPV9 Correlation (B)		Efficacy (Calculated)
	(A)	HPV Fraction Compared to Total Disease (De Martel) [21]	HPV9 Fraction (Harwig) [22]	(A × B)
CIN2+	97.1% (83.5–99.9)	-	82.3%	79.9%
Cervical cancer (CC)	97.4% (85.0–99.9)	100%	89.1% [87.7–90.4]	86.8%
		89.1%		
NIVs	100% (55.5–100)	-	94.4% [91.0–96.9]	94.4%
Vaginal cancer	97.4% (85.0–99.9)	78%	87.1% [78.8–92.6]	66.2%
		67.9%		
Vulvar cancer	97.4% (85.0–99.9)	48%	94.3% [89.1–97.5]	44%
		45.3%		
Penile cancer	100% (−52.1–100.0) ~50%	51%	90.7% [84.1–95.3]	23.15%
		46.3%		
Anal cancer	74.9% (8.8–95.4)	100%	94.4% [89.2–97.5]	70.7%
		94.4%		
Oropharyngeal cancer	77.5% (39.6–93.3)	24%	97.5% [93.7–99.3]	18.11%
		23.4%		
Genital condylomas	99% (96.2–99.9)	-	90%	89.1%

#### 2.1.2. Cost Parameters

The direct costs of the HPV-related health events are taken from Italian published literature. We use the Diagnosis-Related Group (DRG) [23] tariffs for hospitalisations, taking into account the probability of recurrence and, therefore, of readmission. Similarly, official tariffs were used for specialist visits for diagnosis and specific follow-ups for each type of HPV event. In addition, we stratified the use of resources according to the level of severity of the HPV-related condition [24], with particular reference to the staging of tumours [25]. The model therefore considered distinct lifetime costs according to condition severity.

In line with the AIFA price determination for the 9-valent vaccine, the total cost per complete cycle of the 9-valent HPV vaccine was EUR 139.2 (EUR 126 for the acquisition of 2 doses and EUR 13.2 for administration) [26].

With regards to social security costs, a recent study presented at various national and international conferences estimated a social security burden of the diseases under consideration at over EUR 69 million per year [27]. These costs include all disability payments due to the pathologies considered for all those who have paid at least 5 years of employment contributions to the National Social Security Institute (INPS). Of these, about EUR 20.8 million are associated with cancers for which there is a known fraction attributable to HPV. In order to consider these expenditure items within the model, it was assumed that about 90% of women and men diagnosed at working age with one of the cancerous diseases under consideration are eligible for one of the social security benefits (disability allowance). For women who develop moderate to severe cervical intraepithelial neoplasia (CIN), it was assumed that only 50% will claim a social security benefit. The average costs per beneficiary are applied to the fraction of cases that are preventable by vaccination and to the portion considered ‘severe’ (regional and distal forms of ano-genital, head-neck, and cervix cancer). Table 2 shows the direct health and social security costs for each HPV-related disease conditions.

### 2.2. The Herpes Zoster Model

Herpes Zoster (HZ) occurs as a result of the reactivation of a latent Varicella Zoster virus (VZV) infection and its incidence increases with age. In fact, more than two thirds of all cases of HZ occur in patients over 60 years of age, also in the Italian context. While HZ- associated mortality is low, the frequency of its complication, post-herpetic neuralgia (PHN), is about 1 in every 4 cases of HZ. Episodes of PHN can last from several months to years and may require hospitalisation. It has also a significant impact on the patient’s quality of life. The PNPV 2017–19 recommends routine immunization through active offers of HZ vaccination to the cohort of individuals 65 years old, plus the eventual catch-up (maintaining the eligibility and free of charge offer) to all individuals born after 1952 and to individuals aged 50 years and above with risk conditions. In Italy two types of HZ vaccine are currently available: the live attenuated vaccine (ZVL) and the recombinant adjuvanted vaccine (RZV).

The ROI analysis of vaccination against HZ evaluates the following scenarios for the different populations currently eligible for vaccination according to vaccine type (Table 2):For the live attenuated vaccine (ZVL), a non-vaccination scenario was compared with vaccination of the cohort of subjects = 65 years of age at current coverage rates (according to OSMeD data and some regional reports it was estimated to be 20% of the routine cohort) and target coverage (the only cohort for which a target of 50% is set in the PNPV 2017–2019) and of the cohort of subjects aged 50–64 years with risk conditions (current estimated coverage: 5%; assumed target coverage: 20%);For the RZV, in addition to the previous two immunization groups, the cohort of immunocompromised individuals between 18 and 49 years of age with current coverage of 0% and an assumed target coverage of 20% was considered.

The time horizon of the analysis was assumed to be 7 years for both vaccines, in agreement with the horizon set by a similar exercise carried out by the OHE for ZVL (7.6 years) and with a recently published clinical study [28] (Boutry, 2021) for RZV which states that ‘Efficacy against HZ and immune responses to RZV remained high, suggesting that the clinical benefit of RZV in older adults is sustained for at least seven years post-vaccination’.

#### 2.2.1. Epidemiological and Vaccine Efficacy Parameters

The age-stratification of the risk of complications and the immunodepression status was taken from *Istat Multiscopo* data and by using some disease-specific registers and outpatient case-mix systems such as the ACG system used in the Veneto Region [29,30]. Vaccine efficacy is taken from clinical pivotal trials of ZVL and RZV [31,32]. The main epidemiological and clinical parameters used in the HZ model are shown in Table 3.

#### 2.2.2. Cost Parameters

The costs of HZ complications were obtained from a study by Panatto et al. [34], while the costs related to lost productivity were obtained from a recent study by Ruggeri et al. [35]. Some assumptions were necessary in case of lack of data. The cost per ZVL was assumed to be EUR 87.0 per dose while the cost per dose of RZV was assumed to be EUR 166.1 (two doses required) [36] applying the 50% minimum rebate rates for tender products to the public price of these 2 vaccines, after 10% VAT deduction Table 4).

### 2.3. The Influenza Model

Influenza prevention appears particularly important in the elderly (≥65 years) due to their greater frailty with respect to the risk of respiratory complications, but also cardiovascular, renal, and central nervous system complications. The minimum vaccination coverage target recommended by the PNPV 2017–2019 for this population is 75%, while the optimal target is 95%. The ROI model for influenza vaccination includes a vaccination-eligible population of ≥65 years, using 2020 ISTAT Demo data, of 13,693,215 individuals. The population was divided into two groups: low-risk and high-risk elderly, as it is known that people with underlying chronic diseases have a higher risk of developing severe forms of influenza with complications and, consequently, are more likely to be hospitalised. According to the *Passi Argento* study [37], 39.1% of the elderly persons surveyed have no chronic diseases while 60.9% have at least one chronic disease.

In the model, the ROI of influenza vaccination was estimated with the current vaccination coverage compared to the non-vaccination strategy and in the two target coverage scenarios of 75% and 95%. The analysis assumed the use of the quadrivalent inactivated vaccine (QIV), which is currently the most widely used in the Italian context.

The time horizon of the model is one year, as influenza vaccination has to be repeated annually. The model is static, and no attempt was made to include the value of herd immunity.

#### 2.3.1. Epidemiological and Vaccine Efficacy Data

QIV efficacy was assumed to be equal to 61.8% based on an economic evaluation by Capri et al. [38] which compared four influenza vaccines with a non-vaccination option. As regards the influenza attack rate, data from a recent HTA report on the QIV vaccine were used, which estimated an attack rate of 7.2%. According to the *Passi Argento* study, influenza vaccination coverage in those over 65 years of age without chronic diseases is 46.3% while it is 62.9% for those with at least one chronic disease. The type and probability of individual influenza complications were extracted from the study by Meier et al. [39] and are detailed in the appendix. Finally, the probability of hospitalisation in the case of complications in the low- risk elderly was calculated in the same study by Capri et al. as 26.4%, while the probability of hospitalisation in the high-risk elderly was 34.3%.

#### 2.3.2. Cost Parameters

The costs associated with general practitioner (GP) visits for outpatient services are taken from the tariffs provided by the Ministry of Health and amount to EUR 20.66. For hospitalisations the DRG values (published in the *Gazzetta Ufficiale*) were used, except for cardiac and central nervous system complications for which we use a study published by Iannazzo et al. [40] in which the authors estimated that these complications were associated with more than one DRG and calculated a weighted average for the frequency of episodes. Finally, for costs associated with lost productivity, data from the aforementioned recent study by Ruggeri et al. were used (Table 5).

## 3. Results

### 3.1. HPV Model Results

Our model showed that in the absence of HPV vaccination, the total discounted cost due to HPV-related diseases over lifetime was equal to EUR 207 million (EUR 498 million undiscounted) for the NHS and society due to HPV-related diseases (Table 6). Partial coverage rates achieved in 2019 (41.6% in girls and 32.2% in boys) resulted in discounted savings of EUR 43 million (EUR 83 million undiscounted) of this expenditure against EUR 29 million of expenditure of direct and indirect clinical costs for HPV vaccine acquisition costs and vaccine administration costs.

If 2017–2019 PNPV optimal targets were achieved, the savings due to direct and indirect costs would increase up to EUR 104 million (EUR 171 undiscounted) against EUR 74 million invested in the vaccination programme. More than 75% of the economic costs estimated by the model are represented by direct costs (hospitalisations and specialist follow-ups) and, consequently, the largest savings are in these items.

Overall, the return on investment (ROI) is 1.49 at current coverage rates compared to no vaccination and remains positive, with an ROI of 1.40, even at target coverage compared to the non-vaccination strategy. The ROI would be 1.35 when the target vaccination scenario is compared to current vaccination. Note that the undiscounted values would be much more favourable to the vaccination with a ROI of 3.58 for current vaccination compared to no vaccination and an ROI of 2.28 for target vaccination compared to current vaccination.

In a deterministic sensitivity analysis, we varied some parameters that were expected to be of particular importance in estimating the ROI of HPV vaccination: vaccine efficacy (trial confidence interval), lifetime risk of all HPV complications (using some alternative sources), and HPV condition costs (±20%). In all cases examined, the ROI of both current coverage vaccination and the target vaccination scenario in comparison to the non-vaccination strategy remained above 1. Specifically, with regard to the comparison between current vaccination and no vaccination, the ROI varied between 1.06 and 1.75 as the vaccine efficacy varied, between 1.23 and 2.61 as the lifetime risk of HPV-events varied, and between 1.16 and 1.92 as the costs of HPV-conditions varied. As regards the comparison between target vaccination and no vaccination, the ROI varied between 1.04 and 1.42 as the vaccine efficacy varied, between 1.26 and 1.97 as the lifetime risk of HPV-events varied, and between 1.12 and 1.55 as the costs of HPV-conditions varied.

### 3.2. HZ Model Results

Considering vaccination with ZVL, in the case of non-vaccination, the direct costs generated by HZ and its complications would be EUR 57 million, while indirect costs would be over EUR 135 million, for a total of EUR 191 million. The use of ZVL at current coverage rates leads to a reduction of EUR 1.7 million in direct costs and almost EUR 3 million in indirect costs, but the extra cost of vaccination would lead to an ROI of 0.21. A target coverage would increase the savings, but also the cost of vaccination with an ROI of 0.24. Finally, vaccination at target coverage would have a ROI of 0.25 compared to current coverage.

In the case of vaccination with RZV, the direct and indirect costs of HZ and complications in the case of non-vaccination would be EUR 73 million and EUR 164 million, respectively (higher than in the case of vaccination with ZVL because the cohort of immunocompromised individuals is added to the immunisation programme). The use of RZV at current coverage would lead to a reduction of EUR 2.6 million in direct costs and more than EUR 4.5 million in indirect costs, but the higher cost of vaccination would lead to an ROI of 0.08. At target coverage the savings would increase, but so would the cost of vaccination with an ROI of 0.09 (both compared to no vaccination and to current vaccination). The results of the two models are shown in more detail in Table 7 and Table 8.

In general, it can be observed from the two models that indirect costs are significantly higher than direct costs (ratio of 2.5:1), while the total costs generated by HZ are approximately three times those generated by the PHN complication (despite the fact that PHN is associated with higher direct and indirect costs per individual) due to the higher incidence of HZ than PHN.

A sensitivity analysis was performed by varying the parameters with the greatest impact on the ROI of HZ vaccinations, such as of HZ and PHN incidence, vaccine efficacy, and direct and indirect costs. In none of the cases analysed did the ROI approach 1 for either of the two HZ vaccinations considered (data not shown). In a threshold analysis, it was estimated that the incidence of HZ would have to exceed 2.6% for all cohorts considered (0.6% and 0.69% in the base case) to have a ROI greater than 1 for the ZVL vaccine, and even exceed 10% for all cohorts considered for the RZV vaccine. Similarly, indirect costs would have to increase five-fold for the ZVL vaccine and be 15 times higher in the case of the RSV vaccine to reach an ROI equal to 1.

### 3.3. Influenza Model Results

In the case of non-vaccination, the direct costs raised by influenza and its complications would be about EUR 125 million while the indirect costs would be more than EUR 23 million, for a total of EUR 148 million. Compared to non-vaccination, immunization with QIV at current coverage leads to a reduction of EUR 56 million in direct costs and more than EUR 8 million in indirect costs, but the cost of vaccination would lead to an ROI of 0.48. A target coverage of 75% savings would increase to about EUR 65 million in direct costs and almost EUR 11 million in indirect costs, but the increase in vaccination costs would make the ROI compared to non-vaccination strategy equal to 0.48. A 95% of vaccine coverage rate would only slightly increase the ROI to 0.495 but would significantly increase savings associated to disease complications. In contrast, the ROI for vaccination at target coverage of 75% or 95%, compared to current vaccination, was 0.46 and 0.53, respectively. Details of the influenza model results are presented in Table 9.

In the sensitivity analysis we varied the main parameters of the model, using alternative sources for epidemiological data [42,43,44,45] and vaccine efficacy data [46,47] and varying direct and indirect costs. In none of the cases considered did the ROI for influenza vaccine for both target and current coverage exceed 1 compared to non-vaccination. In a threshold analysis, the parameter with the greatest impact on ROI was the attack rate. Doubling this value (from 7.2% in the base case to 14.4%) would increase the ROI close to 1 for current coverage compared to non-vaccination and slightly above 1 for target coverage compared to non-vaccination. Given the uncertainty around this value and the potential underestimation of influenza cases, it is important to take into account the importance of this parameter.

## 4. Discussion and Conclusions

The development of these case studies was inspired by a recent exercise conducted by the UK’s Office of Health Economics (OHE) which had shown similar results, with the ROI of HPV vaccination being the highest and that of Herpes Zoster immunisation the lowest (due to the low incidence of disease). Despite differences in epidemiological, economic, and model structure or assumptions, the OHE results were almost identical to the results of our study with a ROI of 1.5 for HPV vaccination and 0.23 for ZVL vaccination against HZ. The third vaccination considered in the OHE study was pneumococcal vaccination in the paediatric age group, unlike our analysis which included influenza vaccination in the elderly population. The OHE itself is now leading the ongoing discussion between the National Institute for Health and Care Excellence (NICE) and the Joint Council for the Welfare of Immigrants (JCWI) Commission on the most appropriate methodological approach. Specifically, with a Delphi panel, OHE asked experts and senior economists from both bodies to prioritise twelve elements of vaccination and for each of these to assess data gaps and ease of adoption.

Our analysis has some, partly expected, issues to highlight. The HPV vaccination, addressed to an adolescent population, and thus having a long life-expectancy, prevents potentially very serious diseases with consequent high morbidity and mortality, and shows a higher return on investment compared to vaccinations addressed to an elderly population. Moreover, both the HPV vaccine and the influenza vaccine protect not only against an individual risk, like the HZ vaccine, but can potentially reduce the transmission of the infection in the population. However, it must be emphasised that the analysis performed is highly conservative: in fact, as the models are static, they do not directly take into account the potential of this risk of transmission between individuals (herd immunity), resulting in an underestimation of savings in direct and indirect costs of both HPV- and HZ-related complications. In the case of HZ, moreover, since a time horizon of 7 years was used on the basis of clinical follow-up, some advantages in terms of reduction in future cases of HZ and PHN have not been adequately considered, in particular of the RZV vaccine. On the other hand, a sensitivity analysis (data not shown) showed that even doubling the time horizon of the model, the ROI for vaccination with RZV would still be 0.20. Moreover, the exclusion of costs associated to productivity loss due to premature mortality has certainly reduced the potential savings of increasing vaccination coverage, especially for the HPV vaccine.

Another important element to consider is the impact of the discount rate on the results of the analysis. In particular, with regard to HPV vaccination, since savings normally occur over a medium/long time horizon, the value of the undiscounted ROI is almost three times higher than the discounted ROI (or more than twice in the comparison between the scenario with target coverage and non-vaccination). For vaccination against HZ, on the other hand, there is only a small increase in savings (the time horizon being 7 years). The discount rate was not considered for the calculation of the ROI for influenza vaccination given the analysis with a one-year time horizon. Figure 1 and Figure 2 graphically reproduce the impact of the discount rate on the ROI of HPV and HZ vaccination with ZVL or RSV vaccine.

Our analysis also suggests some observations on the importance of indirect costs for the vaccinations considered. While for influenza vaccination the ratio between direct and indirect costs is about 5:1 (essentially due to the fact that an elderly population ≥ 65 years was considered) and for HPV vaccination it is 3:1 (due to the strong impact of hospitalisation costs for the most severe cases of carcinomas), for HPV vaccination the ratio is reversed (approximately 1:2.5). The main reason for this may be the extension of the target population to at-risk individuals aged 50–64 years and, above all, the fact that both HZ and PHN can have a strong impact on the individual’s ability to work for a certain period of time. It should, however, be acknowledged that productivity costs were estimated with a different methodology for HPV vaccination with respect to influenza and HZ vaccination. For HPV vaccination we used social security costs paid to oncology patients while for influenza and HZ vaccination we estimated productivity costs on the basis of days of work lost and average wage rate in Italy using the human capital approach. This choice was due to the different availability of data for the conditions under analysis.

Finally, the three types of vaccination show very different cost impacts: for HPV most of the costs are related to cancer complications rather than to pre-cancerous phases, for HZ the costs are mainly attributable to the acute phase, while for influenza the most important category of costs is that related to complications.

It is important to point out some limitations of the analysis. Firstly, there is uncertainty about some of the data used in the model which have an important impact on the results of the analysis. One example is the attack rate of influenza, which varies greatly depending on the year considered and can strongly influence the ROI results. Another figure that is generally much discussed is the loss of productivity. An attempt has been made to use the most recent sources for the Italian context, but methods and values used to estimate the loss of productivity of a disease are historically a source of discussion.

It should also be emphasised that the major limitation of the ROI approach remains that it does not consider the impact of a technology on the quality of life of the individuals involved. In the case of vaccination against HZ, this appears to be a key issue, given the strong impact of, for example, PHN on patients’ quality of life and the potential long-term duration of this disease. It should also be noted that in contrast to a cost-benefit analysis, in which there is an attempt to quantify the clinical benefits in monetary terms by estimating the willingness to pay of individuals, there is no monetary calculation of clinical benefits in ROI estimation. As a matter of fact, in the last case the change in direct or indirect costs due to the disease are directly quantified in the reduction in hospitalisations or other healthcare resources. The disadvantage of this approach is that it does not directly consider the clinical benefits of a health programme, but the advantage over cost-benefit analysis lies in avoiding the quantitative estimation and ethical difficulties of monetising a gain in quality of life and/or survival.

From the methodological point of view, as mentioned before, a limitation of the analysis is the choice of static models that considers only a fixed cohort of individuals for each model. Dynamic models would have given the chance to estimate the impact of the reduction in disease incidence for future cohorts (for HPV in particular) and the benefits in terms of herd immunity for the population at large (for HPV and influenza). The use of such models goes beyond the scope of our analysis but is more common in cost-effectiveness models. In any case, the use of dynamic models might have increased the number of cases avoided for each disease given by vaccination, and the results of this analysis could be considered conservative. On the other hand, in the case of the HPV vaccine, the basic reproduction number of HPV is generally marginal, and the benefits of a very high target of coverage could be overestimated using a linear assumption, as in our study.

More generally, it is evident that, despite the fact that the use of ROI in decision-making aims to include some elements that are more difficult to incorporate in a classic cost-utility analysis (which in most cases is conducted using the NHS perspective, thus explicitly excluding indirect costs), the economic and social value of vaccinations might be underestimated even in this case. Indeed, it remains complicated (and not included in our models) to quantify, for example, avoided costs in terms of lost productivity due to premature death or avoided costs on the part of caregivers, or some important benefits in terms of reduced transmission of infection (as for HPV or influenza, in our models) or other less intuitive aspects such as the potential reduction in antibiotic resistance.

In conclusion, our study is an exercise that compares the ROI of different vaccine programmes in the Italian context and aims to offer a ground for discussion and valorisation of the vaccine ‘basket’ within the expenditure for prevention by highlighting the different peculiarities of some vaccine strategies that must be taken into account in decision-making.

## Figures and Tables

**Figure 1 vaccines-11-00924-f001:**
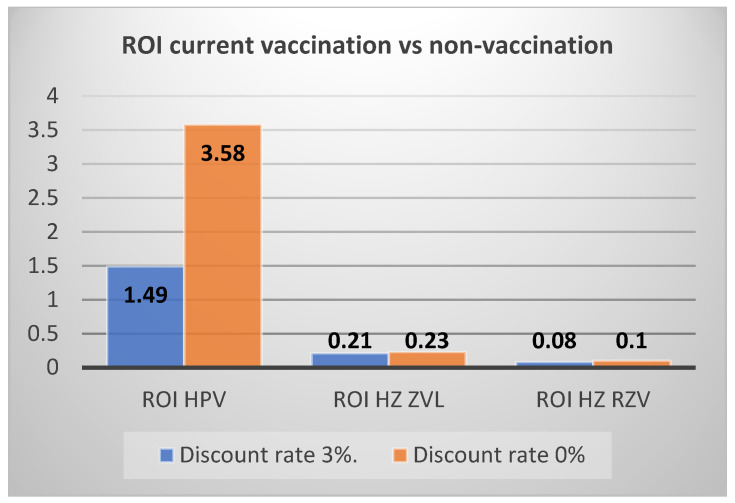
ROI with and without discount rate for HPV and HZ vaccinations (current vs. no vaccination).

**Figure 2 vaccines-11-00924-f002:**
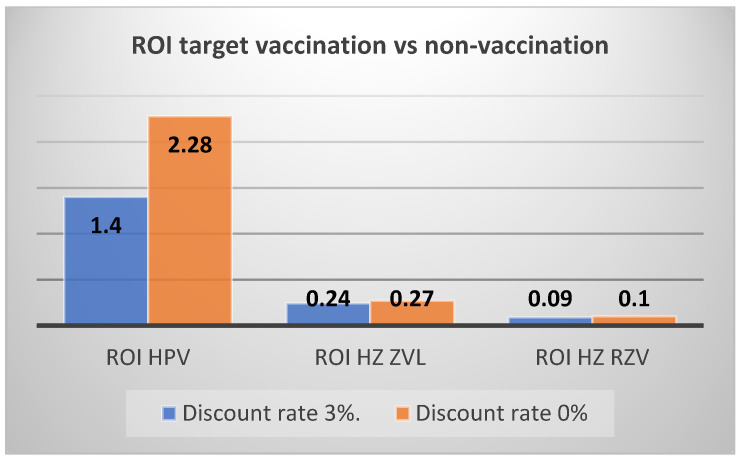
ROI with and without discount rate for HPV and HZ vaccinations (target vs. no vaccination).

**Table 2 vaccines-11-00924-t002:** Direct costs and social security costs for HPV-related diseases.

Category of Costs	Total Direct Cost (EUR) (Females) [24]	Total Direct Costs (EUR) (Males) [24]	Social Security Costs (EUR) [27]
CIN1	452	-	Not applicable (na)
CIN2	1485	-	4265
CIN3	1972	-	4265
Cervical cancer, local disease	20,653	-	na
Cervical cancer, regional disease	35,930	-	9131
Cervical cancer, distant disease	34,575	-	9131
Vaginal cancer, local disease	7703	-	na
Vaginal cancer, regional disease	19,836	-	8150
Vaginal cancer, distant disease	29,647	-	8150
Vulvar cancer, local disease	7184	-	na
Vulvar cancer, regional disease	16,033	-	8150
Vulvar cancer, distant disease	20,365	-	8150
Penile cancer	-	10,498	8218
Anal cancer, local disease	9812	9812	na
Anal cancer, regional disease	18,480	18,480	9136
Anal cancer, distant disease	11,994	11,994	9136
Head & Neck, local disease	10,082	10,082	na
Head & Neck, regional and distant disease	28,572	28,572	9308
Genital warts	700	495	na
Recurrent respiratory papillomatosis	195,815	195,815	9308

**Table 3 vaccines-11-00924-t003:** Epidemiological and clinical parameters for the HZ model.

	Immunocompromised Adults		Adults with Risk Conditions		Routine Cohort		Notes
	18–49 Years		50–64 Years		65 Years		
Healthy people	-	-	-	-	38.7%	240,903	Healthy people from multiscope 2019
With at least one risk condition and immunocompetent	-	-	32.1%	4,364,595	45.0%	280,120	Data from Veneto Region [29,30]
Immunocompromised	1.4%	334,455	4.3%	587,611	16.3%	101,466	Veneto Region ACG system report: 18–49 y (4.34%)–50–64 (4.32%) > 65 (16.3%) [29,30]
HZ Incidence	0.01596	RR Immunocompromessi = 3	0.006916	RR a rischio = 1.3	0.6070%		Alicino 2017 [33]
PHN incidence	0.002676492		0.001159813		0.001465298		
Current coverage rate	0%		5%		20%		Assumptions from Osmed data
Target coverage rate	20%		20%		50%		Assumption
VE ZVL (1 dose)	na		HZ: 64% PHN: 63%		HZ: 64% PHN: 66%		RCP ZVL and STIKO report
VE RZV (2 dose)	HZ: 71.80% PHN: 89%		At risk: HZ: 96.6%, PHN: 100% Immunocompromised HZ: 67.3%, PHN: 88%		healthy: HZ: 97.4%, PHN: 100%; At risk: HZ: 96.6%, PHN: 100% Immunocompromised HZ: 67.3%, PHN: 88%		RCP EMA RZV and STIKO report

**Table 4 vaccines-11-00924-t004:** Cost parameters for HZ model.

Cost Category	Value (EUR)	Source
ZVL acquisition cost	87.00	Farmadati [26]
RZV acquisition cost	166.10 per dose (2 doses)	Farmadati [26]
Administration cost	6.64	Coretti et al., 2016 [36]
HZ treated as outpatient	145.01	Panatto et al., 2015 [34]
HZ hospitalised	3063	Panatto et al., 2015 [34]
PHN treated as outpatient	661.92	Panatto et al., 2015 [34]
PHN hospitalised	3316	Panatto et al., 2015 [34]
Daily productivity loss	129.60	Calculated from Ruggeri et al., 2020 [35]
**Other parameters**	**Value**	
Percentage of hospitalised HZ	1.65%	Panatto et al., 2015 [34]
Percentage of hospitalised PHN	4.06%	Panatto et al., 2015 [34]
Productivity loss HZ (days)	6	Ruggeri et al., 2020 [35]
Productivity loss PHN (days)	10	Ruggeri et al., 2020 [35]
Percentage of workers (65 years or more) Percentage of workers (50–64 years)	30% 61.2%	Assumption ISTAT DEMO [10]

**Table 5 vaccines-11-00924-t005:** Cost parameters for the influenza model.

Cost Category	Cost (EUR)	Notes
QIV acquisition cost	11.08	Ex-factory price
Administration cost	6.16	Nomenclatore tariffario delle prestazioni aggiuntive, Accordo Collettivo Nazionale medici generici, 23 marzo 2005 [41]
GP visit	20.66	Tariffario prestazioni ambulatoriali 2013, Ministero della Salute [21]
Bronchitis (hospitalisation)	1832	DRG 097, bronchitis and asthma, age > 17 years without CC [21]
Pneumonia (hospitalisation)	3558	DRG 089 simple pneumonia, age > 17 years with CC [21]
Unspecified respiratory tract infection	4422	DRG 080, respiratory infections and inflammations, age > 17 years without CC
Cardiac complications	3544	Iannazzo, 2011 [40]
Renal complications	3734	DRG 316, renal failure [21]
Central nervous system complications	3507	Iannazzo, 2011 [40]
Otitis media	1247	DRG 069, otitis media and upper respiratory tract infections, age > 17 years without CC [21]
Gastrointestinal haemorrhage	2091	DRG 175, gastrointestinal haemorrhage without CC [21]
Productivity loss due to influenza	4.7 days	Ruggeri et al., 2020 [35]
Daily wage	103.03	Ruggeri et al., 2020 [35]
Percentage of workers ≥ 65 years	4.87%	Calculated from WHO report

**Table 6 vaccines-11-00924-t006:** ROI results (HPV model).

	No Vaccination	Current Vaccination	Target Vaccination
Vaccine cost	EUR 0	EUR 28,713,084	EUR 73,560,963
Direct costs of HPV-related diseases	EUR 154,078,832	EUR 121,320,045	EUR 74,483,857
Indirect costs of HPV-related diseases	EUR 53,140,903	EUR 43,111,943	EUR 29,488,385
Total costs	EUR 207,219,735	EUR 193,145,073	EUR 177,533,205
ROI vs. no vaccination		1.49	1.40
ROI vs. current vaccination			1.35

**Table 7 vaccines-11-00924-t007:** ZVL results.

	No Vaccination	Current Vaccination	Target Vaccination
Vaccine costs	EUR 0	EUR 19,831,988	EUR 110,885,070
Direct costs of HZ/PHN complications	EUR 56,517,312	EUR 54,814,827	EUR 47,064,976
Indirect costs of HZ/PHN complications	EUR 134,518,156	EUR 131,632,086	EUR 114,466,618
Total costs	EUR 191,035,467	EUR 206,278,901	EUR 272,416,664
ROI vs. no vaccination		0.21	0.24
ROI vs. current vaccination			0.25

**Table 8 vaccines-11-00924-t008:** RSV results.

	No Vaccination	Current Vaccination	Target Vaccination
Vaccine costs	EUR 0	EUR 83,960,127	EUR 484,266,536
Direct costs of HZ/PHN complications	EUR 73,064,973	EUR 70,438,608	EUR 57,261,427
Indirect costs of HZ/PHN complications	EUR 163,772,122	EUR 159,232,906	EUR 130,804,974
Total costs	EUR 236,837,095	EUR 313,631,641	EUR 672,332,937
ROI vs. no vaccination		0.08	0.09
ROI vs. current vaccination			0.09

**Table 9 vaccines-11-00924-t009:** Influenza model results (at 75% and 95% coverage targets).

	No Vaccination	Current Vaccination	Vaccination at 75% Target	Vaccination at 95% Target
Vaccine costs	EUR 0.00	EUR 133,166,249	EUR 159,657,432	EUR 178,118,186
Direct costs of influenza (no complications)	EUR 7,862,407	EUR 4,456,558	EUR 4,435,542	EUR 4,078,589
Direct costs of complications	EUR 117,071,491	EUR 64,209,048	EUR 54,765,739	EUR 46,307,002
Indirect costs	EUR 23,250,293	EUR 15,145,006	EUR 12,473,782	EUR 9,600,046
Total costs	EUR 148,184,192	EUR 216,976,863	EUR 231,332,497	EUR 238,103,824
ROI vs. no vaccination		0.48	0.48	0.495
ROI vs. current vaccination			0.46	0.53

## Data Availability

Not applicable.

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
