# Peer review of "Return on Investment (ROI) of Three Vaccination Programmes in Italy: HPV at 12 Years, Herpes Zoster in Adults, and Influenza in the Elderly"

_vaccines, 2023, doi:10.3390/vaccines11050924_

Round 1

Reviewer 1 Report

General Remarks

I have reviewed this manuscript before and, as far as I can tell, the manuscript has not changed since then. Needless to say, none of my suggestions appear to have been considered, including the advice to have the manuscript edited for English language. As arbitrary example: “Vaccination is routinary offered …” (line 139; should be routinely). 

In my first review, I strongly criticized the linear approach to vaccine effects. Rereading the (largely identical) discussion section I have to admit that the problem is acknowledged there, at least to some extent. Even though I still think the analysis, especially for HPV, lacks validity and may grossely understimate the effect of a modest vaccination uptake among 12-year olds I do recognize the fact that this is the approach generally chosen in health economic analyses of vaccines. 

I still recomment careful grooming of the manuscript for careless errors (e.g. “Valore” in Table 4) and language. Why are there yellow highlights in the manuscript?

Author Response

Dear reviewer,

thanks for your comments and suggestions. We have now edited some english errors, hopefully there are not other mistakes. We perfectly understand you point on the need for a dynamic model, but this was beyond the scope of the analysis. We have tried to acknowledge this as a limitation in the discussion.

Thanks

Reviewer 2 Report

The authors addressed most of my minor comments but failed to adequately address my major concerns about the chosen methodology. I don't see any clear justification of why different evaluation methods have been used for different vaccines, and why the cost-of-illness approach does not include all the benefits that are typically included. 

Author Response

Dear reviewer, 

thanks again for your comments. 

We have now acknowledged in the discussion the use of different approaches to estimate indirect costs for HPV vs flu and HZ vaccinations, and tried to explain the reasons for this choice. 

Hope this is fine

regards

Round 2

Reviewer 2 Report

The authors have addressed my comments.